# New Caregiver Diagnoses of Severe Depression and Child Asthma Controller Medication Adherence

**DOI:** 10.3390/ijerph20115986

**Published:** 2023-05-29

**Authors:** Janet M. Currie, Michele Mercer, Russ Michael, Daniel Pichardo

**Affiliations:** 1Center for Health Wellbeing, 185A JRR Building, Princeton University, Princeton, NJ 08540, USA; 2Blue Health Intelligence®, Chicago, IL 60601, USA

**Keywords:** childhood asthma, caregiver depression, caregiver diabetes

## Abstract

Background and Objectives: Children with asthma who have depressed caregivers are known to be less adherent to medication regimes. However, it is less clear how adherence responds to a caregiver’s new diagnosis of severe depression or whether there is a similar relationship with other serious caregiver diagnoses. The hypothesis is that adherence worsens both with new diagnoses of depression and possibly with new diagnoses of other serious conditions. Methods: This study follows a cohort of 341,444 continuously insured children with asthma before and after a caregiver’s new diagnosis of severe depression or another serious health condition. The effect of a new depression diagnosis on a child’s medication adherence is compared to the effect of new diagnoses of other common caregiver chronic conditions including diabetes, cancer, congestive heart failure, coronary artery disease, and chronic obstructive pulmonary disease. Results: Results show that children’s medication adherence declines following a caregiver’s new diagnosis of severe depression, but that it also declines following a caregiver’s new diagnosis of diabetes. There is no association with new diagnoses of the other caregiver chronic conditions examined. Conclusions: Children whose caregivers have a new diagnosis of depression or diabetes may be at increased risk of deterioration in their medication adherence. These caregivers may benefit from additional support and follow-up. The relationship between caregivers’ health and children’s medication adherence is complex and deserves further study.

## 1. Introduction

Asthma affects 7% of all children younger than 18, making it one of the most common childhood chronic conditions [1]. Recent meta-analyses confirm that caregiver depression is linked to childhood asthma [2,3], with some studies finding rates of maternal depression that are more than twice as high among children with asthma [4]. There are many possible reasons for such an association. For example, children in families with lower socioeconomic status are more likely to suffer ill health [5] and poverty is also a risk factor for adult depression [6]. Parental hospitalization has been shown to have severe economic consequences for many households [7]. Severe child illness can affect the stability of the household as well as the parents’ capacity to work [8,9].

It is possible, however, that parental depression directly either causes asthma or impedes a caregiver’s ability to manage the child’s illness, leading to an exacerbation of symptoms. Several studies of these issues are reviewed in Appendix A. Two large-scale birth cohort studies suggest that caregiver depression itself leads to the development of asthma in the child, perhaps by elevating the child’s stress levels [10,11].

Other studies examine the relationship between a caregiver’s depression and the extent to which a child’s asthma is controlled. For example, one cross-sectional study of 177 minority children with asthma found that mothers with more depressive symptoms reported more problems with medication adherence and less confidence in the ability of medications to control the child’s asthma [12]. A similar study of 78 children found that depressive symptoms were associated with worse control of a child’s asthma [13]. A difficulty with cross-sectional studies such as these is that the causality could be reversed—for example, mothers who have more difficulty managing a child’s asthma may become depressed.

Another study of 204 inner-city children recruited after emergency room visits for asthma followed them over time and found that increases in self-reported maternal depression were associated with reductions in self-reported medication compliance [14]. Notably, this study found that changes in a caregiver’s depression status were more important than baseline levels. A second study that followed children via phone surveys found that maternal depressive symptoms at baseline predicted children’s asthma symptoms six months later [15].

Since 18.5% of U.S. adults experienced symptoms of depression in 2019 [16], more research into the mechanisms linking these two illnesses and examines the impact of new diagnoses is necessary if morbidity from childhood asthma is to be effectively addressed. Ideally such studies would have larger sample sizes as well as cohort data that follows the same child over time.

This study follows a national cohort of children who were covered by private health insurance for at least one year between June 2006 and June 2018. In order to see whether there is something unique about caregiver depression, the effect of a new severe depression diagnosis is compared to that of a new diagnosis for other serious conditions including cancer, diabetes, congestive heart failure (CHF) or coronary artery disease (CAD), and chronic obstructive pulmonary disease (COPD). This study differs from previous work because the data follow the same child before and after the caregiver’s diagnosis in order to observe changes in medication compliance over time rather than only observing a correlation between medication adherence and the parent’s mental health condition.

## 2. Methods

This study used administrative claims from Blue Health Intelligence (BHI), a licensee of the Blue Cross Blue Shield Association. The dataset includes enrollment information, medical claims, and pharmacy claims for Blue Cross and Blue Shield plan beneficiaries across the country. It does not include Medicaid or Medicare Advantage. All children who meet the study criteria described below are included.

The BHI dataset contains de-identified data for more than 48.4 million unique individuals enrolled in Blue Cross Blue Shield commercial health plans. This study uses only de-identified data from a national administrative claims warehouse; therefore, it is not human subjects research and no ethical approval was required. Data use agreements between BHI and the originating Blue Cross Blue Shield insurers impose restrictions on use, such as using the data within the secure warehouse environment, which were followed.

The primary insurance holder is treated as the first caregiver and the secondary adult (if there is one) as a second caregiver. All dependents under 18 years old were considered children. The study includes all children between the ages of 2 and 11 years old who were observed to have one of the asthma diagnoses listed in Appendix A and subsequent treatment with one of the medications listed in Appendix A, and who had at least one year of continuous enrollment prior to and following the first observed asthma claim. The study focuses on this age group because older children may have started to assume more responsibility for their own asthma management and thus may be less dependent on a caregiver to monitor their adherence to medication regimes. Any child with family members (including siblings) having other (non-asthma) serious chronic conditions at baseline was removed from consideration in order to ensure that families had a similar initial level of overall health and disease burden prior to the caregiver’s new diagnosis of severe depression or other serious health condition. The coding of conditions was based on the Department of Health and Human Services Hierarchical Condition Categories (HSS-HCC), which was developed after the passage of the Affordable Care Act to pay health insurers in the ACA marketplace [17].

These steps left 341,444 children who had both an asthma diagnosis and subsequent treatment. The average child was in the study for 63 months. Because all of the children in these data had private health insurance, this study does not focus on lack of insurance coverage as a reason for differences in medication adherence, but rather on risk factors among insured children.

Of these 341,444 children, the study focused on those who had demonstrated some previous reliance on asthma medications. Children with 30 days or more of medication coverage in the previous quarter were considered to have shown previous reliance on asthma medications. There were 156,209 children who met this criterion. A full list of included asthma medications appears in Appendix A. A day is considered covered if it was within the number of days supplied in the previous medication fill.

The measure of medication adherence is the medication possession ratio, which is equal to the number of days supply (of one of the listed medications) divided by the number of days in the period. This variable is defined with reference to calendar quarters. This measure is commonly employed by researchers using administrative pharmacy claims databases [18]. When using pharmacy claims the best one can say is that the subscription was filled and that the patient had enough supply to cover the days in the quarter. This is a minimum criterion for medication compliance, but whether the child actually took the medication as prescribed is not observed in these data.

A new caregiver diagnosis of severe depression was defined as the first observed instance of one of the depression diagnoses that are included in the HSS-HCC model with a modifier of “severe”. These codes are F322, F323, F332, and F333 which cover severe major depressive disorders with and without psychotic features.

To control for other determinants of the child’s asthma medication compliance, the estimation models included indicators for the child’s age, gender, and number of siblings. In the main model, social determinants of health were accounted for by including the zip code’s social vulnerability index (SVI), which was developed by researchers at the Centers for Disease Control using data from the American Community Survey, the Centers for Disease Control, and the Agency for Healthcare Research and Quality to identify populations most at risk due from external stresses on human health [19].

Since rural residence may be a significant determinant of medication access, an indicator for urban/rural classification was also included. It is based on the rural–urban commuting area scores developed by the U.S. Department of Agriculture Food Access, Rural Health Research Center, and the Health Resources and Services Administration [20].

In alternative models, the SVI’s individual zip-code level components are included rather than the SVI. The components include the percentage poor, the unemployment rate, median per capita income, the percentage without a high school diploma, the percentage of the population 65 or older, the percentage of the population aged 17 or younger, the percentage with a disability, the percentage of single-parent households, the percentage minority, the percentage of people aged 5 or older who speak English less than well, the percentage of households in mobile homes, the percentage of households in housing with greater than 10 units, the percentage of households experiencing crowding (i.e., more people than rooms), the percentage in group quarters, and the percentage uninsured. The components of the SVI may be highly correlated with each other which is why specifications with the overall index may be preferred.

The statistical analysis takes advantage of the fact that a cohort of children is followed before and after their parent’s new diagnosis of a behavioral health condition. Specifically, linear fixed effects models of the effect of a caregiver’s new severe depression diagnosis on the child’s number of days’ supply of controller medications within each 90-day period are estimated controlling for the variables described above. The mean number of quarters observed per diagnosis is 8.2 and the mean number of quarters observed post-diagnosis is 2.2. By estimating a model including child fixed effects (which can be thought of as a unique zero–one indicator for each child), these models effectively look at the change in medication refills that takes place after a parent is diagnosed. Fixed effects models offer a powerful way to control for any observed or unobserved characteristic of the child and family that is constant over time such as mean household income over the time period that the household is observed [21]. The child fixed effect also captures some observable characteristics such as caregiver and child age at the start of the observation window. The coefficient on child age can then be interpreted as the effect of changing age. Moreover, since the change in the child’s age is the same as the change in the parent’s age, the latter is not included.

To see if new diagnoses of other common caregiver health conditions have a similar effect, the analysis is repeated for caregivers with a new diagnosis of cancer (all HSS-HCC codes involving neoplasms), diabetes mellitus, congestive heart failure (CHF) or coronary artery disease (CAD), and chronic obstructive pulmonary disease (COPD). 

All analyses were performed with R 4.0.2 (R Core Team, 2020) using the plm package [22]. The reporting of this study follows the STROBE guidelines for the reporting of cohort studies, which is a checklist for insuring that the main points are properly described [23].

## 3. Results

Table 1 shows the means and standard deviations of the control variables for the 619 children whose caregivers had a new diagnosis of severe depression compared to the 155,590 children whose caregivers did not. The table also reports the difference between the means for the children with and without caregivers with depression diagnoses, the *p*-value for the difference (based on Chi-squared tests), and the upper and lower bounds of a 95% confidence interval. Table 1 shows that families whose caregivers have new diagnoses of depression are different from other families in ways that might also affect their children’s medication compliance. The table suggests that children with affected caregivers are younger and have fewer siblings. They also have slightly lower per capita income and are more likely to live in a zip code with more children, more single-parent households, and more households without vehicles. These differences all emphasize the importance of being able to follow the same child over time as in this study so that differences between households with and without caregivers with depression can be controlled.

Table 2 shows the estimated impact of a new caregiver diagnosis of severe depression on medication adherence. That is, the table shows the results of estimating regression models in which medication adherence is the dependent variable and the independent variables include child age, an indicator equal to one if the child had full medication coverage in the previous calendar quarter and zero otherwise, the number of siblings, an indicator equal to one if the postal code is urban and zero otherwise, the SVI, which is standardized to have a mean of zero and a standard deviation of one, and a zero–one indicator for each child (i.e., the child fixed effect).

Table 2 shows that a new diagnosis is associated with a statistically significant reduction of 1.76 days in medication coverage (based on a *t*-test). This effect is comparable to gaining an additional half year of child age, gaining an additional sibling in the household, or a one standard deviation change in the SVI. Residence in an urban zip code has an effect roughly twice as large. Estimates controlling for components of the SVI rather than SVI are shown in Appendix A. The estimated effects of a new diagnosis of severe depression are very similar. The only individual components of the zip code SVI that are individually statistically significant are per capita household income and the percentage of adult residents without a high school degree, but these zip-code level attributes have relatively small effects. As discussed above, it may be more meaningful to focus on the overall SVI index rather than the individual components.

Table 3 examines the impact of new diagnoses of other common caregiver chronic conditions. The models estimated are identical to those in Panel A of Table 2 except for the chronic condition indicator. The estimates indicate that of the additional conditions considered, only a new caregiver diagnosis of diabetes has a statistically significant effect.

Figure 1 allows a visual comparison of the estimated effects. The impact of a new diabetes diagnosis is shown to be larger than that of a new severe depression diagnosis. This result suggests that although a caregiver’s new severe depression diagnosis has the potential to disrupt a child’s medication adherence, some other new diagnoses of chronic conditions may also have this effect.

## 4. Discussion

This study shows that in a cohort of children who use asthma controller medications, a new parental diagnosis of severe depression or diabetes is followed by a deterioration in medication adherence.

There are many reasons that a new diagnosis of severe depression in a caregiver might be associated with reduced medication adherence among children. The caregiver might find that the new demands associated with their own treatment (such as additional therapy) crowd out the time available for dealing with their child’s medical problems. Or they might find the diagnosis itself overwhelming so they have more difficulty dealing with other demands on their attention. The depression itself might also make it more difficult for the caregiver to attend to the needs of the child.

An alternative, non-causal explanation for an association is that some caregivers may be both more susceptible to parental mental health problems and less likely to be fully compliant with their child’s asthma medication regime. This analysis controls for this possibility by following the same child over time and focusing on whether a caregiver’s new diagnosis of severe depression changes the child’s medication adherence. It is also possible that in some cases a caregiver’s new diagnosis of depression affects medication adherence by disrupting the child’s insurance coverage (if for example, the caregiver was unable to work). However, by focusing on a population that is continuously insured, that possibility is eliminated in this analysis.

The strength of the analysis stems from the comprehensive nature of the underlying database, which allows a large group of children who fit the study criteria to be studied. Further strengths stem from the fact that children can be followed over time in order to focus on those whose caregivers receive a new diagnosis of severe depression. Including a child fixed effect and focusing on changes in adherence that take place with changes in diagnoses helps to control for a large array of potential confounders.

Finally, this rich data can be exploited in order to compare the impact of a new caregiver diagnosis of severe depression with the impact of new caregiver diagnoses of other serious health conditions. This analysis reveals that caregiver depression may not be unique in inducing problems with child medication compliance since an effect is also found for a new diabetes diagnosis. This finding raises the question of why no effect is found for a new cancer, COPD, or CHF diagnosis, especially since there are relatively large numbers of caregivers in these data who have new COPD and cancer diagnoses. It is possible that there is initially more social support (and less stigma) for people with these new cancer, COPD, or CHF diagnoses compared to new diagnoses of depression [24] or diabetes [25], and that social support is important in mediating the relationship between new caregiver diagnoses and child outcomes. 

## 5. Limitations

A limitation common to many studies of medication adherence is that the measure of adherence captures whether prescriptions were filled in a timely way but does not capture whether children actually used the medication. In addition, many asthma medications are inhaled and fills of inhaled medications can be particularly difficult to connect to daily use. With a relatively small number of children affected by a new diagnosis of caregiver severe depression, the study also lacks the statistical power to examine the impact of the changes in medication compliance on subsequent child health outcomes such as emergency room visits. A further limitation is that the components of the SVI are not available at the level of the individual family, that is, family-level income and education are not observed in these claims data. Moreover, it is important to keep in mind that all of the children in this study were privately insured, so the generalizability of the findings to families who are publicly insured requires further research.

Finally, it is possible that a new caregiver condition began to affect family functioning prior to the date that the condition was recorded in the claims data. In this case, fixed effects estimates will tend to understate the effect of a new diagnosis because some of the comparison period before the diagnosis was actually affected.

## 6. Conclusions

The results of this study indicate that a new caregiver diagnosis of severe depression reduces a child’s medication use by about two days per calendar quarter on average. Since it is possible that a caregiver being diagnosed with any serious new health condition could disrupt a child’s medication adherence, a range of other relatively common caregiver diagnoses were also examined. The results show that a new caregiver diagnosis of diabetes actually has a larger significant effect than a new diagnosis of severe depression. This result suggests that although a new caregiver diagnosis of severe depression can reduce medication adherence among children with asthma, other new caregiver diagnoses, particularly diabetes, could also have this effect.

These results suggest that it can be beneficial for clinicians and care managers treating children to understand the dynamics associated with caregiver illness in order to implement targeted intervention strategies for the child. Children of caregivers who are newly diagnosed with severe depression or diabetes may require more caregiver support and more frequent follow-up than families without these challenges, particularly given the stigma that can be associated with these diseases [3,24,25]. Simplifying medication regimes by minimizing the number and frequency of medications and making them easily accessible (e.g., through home delivery or school nurses) could also be helpful. At least one study suggests that treating mothers for depression may improve children’s asthma symptoms [26]. Further research to identify children at risk, as well as helpful interventions, would be useful.


**Article Summary**


New caregiver diagnoses of severe depression predict declines in medication adherence among children with asthma. Declines are compared to those for other new caregiver diagnoses.


**What is Known on This Subject**


Children with asthma are less likely to adhere to medication regimes when their caregivers are depressed. Mothers of children with asthma are also more likely to be depressed. However, it is unclear whether caregiver depression triggers non-adherence.


**What This Study Adds**


We follow a cohort of children with asthma before and after a caregiver’s new diagnosis of severe depression. The effect on medication adherence is compared to the effect of new diagnoses of other serious caregiver chronic conditions.

## Figures and Tables

**Figure 1 ijerph-20-05986-f001:**
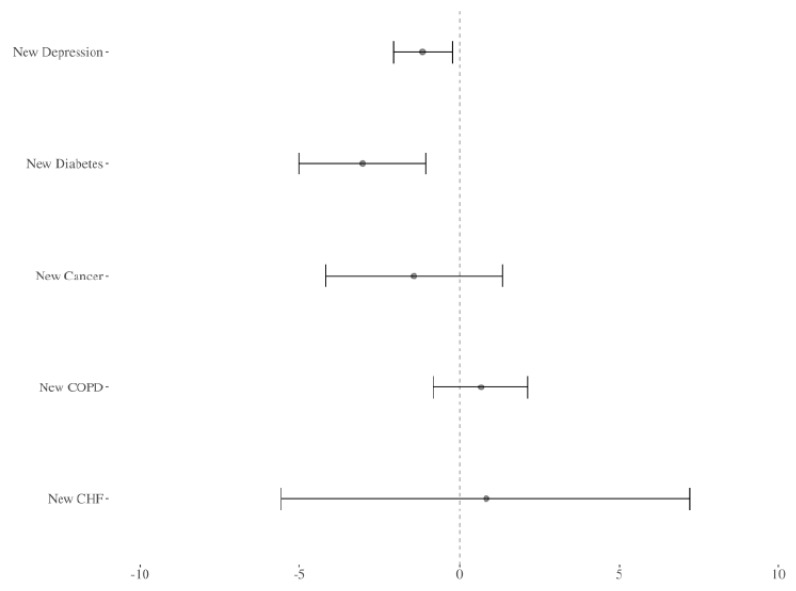
Comparison of the impacts of new severe depression diagnoses and other new caregiver diagnoses on the medication compliance of asthmatic children. The figure shows the point estimates and a 95% confidence interval for diagnoses of severe depression (from Table 2) and other new caregiver chronic conditions (all panels of Table 3). COPD stands for chronic obstructive pulmonary disease and CHF stands for congestive heart failure.

**Table 1 ijerph-20-05986-t001:** Means *±* standard deviations of control variables for children with a 30-day supply of controller medications in the previous quarter.

	New Parent Depression Diagnosis	No New Parent Depression Diagnosis	Difference [2] Minus [1]	*p*-Value for Diff	Lower CL	Upper CL
Number Children	619	155,590				
Child Age	4.84 ± 2.39	6.04 ± 2.89	−1.198	0.000	−1.387	−1.008
Child Gender	388 (63%)	97,066 (62.4)	0.002	0.938	−0.037	0.041
Sibling Count	1.15 ± 0.98	1.23 ± 1.00	−0.082	0.038	−0.159	−0.005
*Variables Measured at the Zipcode Level*						
Zip Code is Urban	524 (85%)	130,073 (84%)	0.010	0.536	−0.019	0.039
Social Vulnerability Index	0.54 ± 0.25	0.53 ± 0.24	0.009	0.343	−0.010	0.029
*Components of the SVI*						
Poverty Rate	12.04 ± 7.46	11.91 ± 7.86	0.131	0.665	−0.463	0.725
Unemployment Rate	7.17 ± 3.38	7.00 ± 3.36	0.167	0.223	−0.102	0.436
Per Capita Income, 1000 s	30.95 ± 10.54	32.10 ± 12.35	−1.146	0.007	−1.985	−0.307
Percentage Without High School Diploma	9.95 ± 6.61	10.16 ± 7.23	−0.205	0.443	−0.731	0.320
Percentage Population Aged 65 or Older	13.93 ± 5.24	14.4 ± 5.21	−0.469	0.027	−0.886	−0.053
Percentage Population Age 17 or Younger	24.16 ± 4.42	23.72 ± 4.64	0.441	0.014	0.090	0.793
Percentage Disabled	11.78 ± 4.03	11.87 ± 4.45	−0.089	0.587	−0.409	0.232
Percentage Single-Parent Households	8.76 ± 3.62	8.43 ± 3.74	0.332	0.024	0.044	0.620
Percentage Minority	27.33 ± 21.87	27.15 ± 22.37	0.172	0.846	−1.568	1.912
Percentage Not Fluent in English	2.35 ± 3.75	2.53 ± 3.96	−0.179	0.239	−0.478	0.119
Percentage in Housing with >10 Units	8.96 ± 8.72	9.53 ± 10.7	−0.570	0.107	−1.264	0.124
Percentage in Mobile Homes	5.6 ± 7.59	6.1 ± 8.42	−0.501	0.104	−1.105	0.103
Percentage Crowded Dwellings	2.06 ± 2.12	2.11 ± 2.42	−0.051	0.554	−0.220	0.118
Percentage Households with No Vehicle	5.49 ± 4.14	5.89 ± 5.7	−0.397	0.018	−0.727	−0.067
Percentage in Group Quarters	1.79 ± 3.42	1.97 ± 4.16	−0.180	0.195	−0.453	0.092
Percentage Uninsured	10.35 ± 5.68	10.38 ± 6.27	−0.036	0.875	−0.488	0.416

Notes: There are 1,101,657 quarters of child data, or approximately 7.05 quarters per child. CL stands for confidence limit, and the upper and lower CLs given are for a 95% confidence interval.

**Table 2 ijerph-20-05986-t002:** Child fixed effects models of the effects of a caregiver’s new diagnosis of severe depression on medication compliance for children with 30+ days supply.

	Coefficient Estimate	Standard Error	*p*-Value	Lower CL	Upper CL
New Depression Diagnosis	−1.761	0.786	0.025	−3.050	−0.473
Full Coverage Previous Quarter	14.114	0.088	0.000	13.969	14.259
Child Age	−3.445	0.017	0.000	−3.473	−3.418
Sibling Count	−1.544	0.114	0.000	−1.730	−1.357
Social Vulnerability Index	−1.277	1.007	0.205	−2.929	0.374
Zip Code is Urban	−2.990	1.092	0.006	−4.781	−1.199

Notes: There are 1,101,657 quarters of child data, or approximately 7.05 quarters per child. Coefficient estimates are from a child fixed effects model that also included all of the variables shown in the table. Confidence limits (CL) are for a 95% confidence interval.

**Table 3 ijerph-20-05986-t003:** Child fixed effects models of the effects of a new chronic condition diagnoses on medication compliance in children with a 30+ days supply of controller medication.

	[1]	[2]	[3]	[4]	[5]
	Coefficient Estimate	Standard Error	*p*-Value	Lower CL	Upper CL
*Panel A: New Diabetes Diagnosis (n = 3638)*					
New Diabetes	−3.018	1.210	0.013	−5.002	−1.034
Full Coverage Previous Quarter	27.716	0.197	0.000	27.393	28.040
Child Age	−0.042	0.026	0.109	−0.084	0.001
Sibling Count	−1.338	0.074	0.000	−1.460	−1.216
Social Vulnerability Index	−4.343	0.308	0.000	−4.848	−3.837
Zip Code is Urban	−2.401	0.205	0.000	−2.737	−2.064
*Panel B: New Chronic Obstructive Pulmonary Disease Diagnosis (n = 6958)*					
New COPD Health	0.663	0.895	0.459	−0.805	2.131
Full Coverage Previous Quarter	27.928	0.198	0.000	27.604	28.252
Child Age	−0.036	0.026	0.167	−0.078	0.007
Sibling Count	−1.355	0.074	0.000	−1.477	−1.234
Social Vulnerability Index	−4.349	0.308	0.000	−4.855	−3.844
Zip Code is Urban	−2.307	0.205	0.000	−2.643	−1.970
*Panel C: New Cancer Diagnosis (n = 1863)*					
New Cancer DX	−1.410	1.685	0.403	−4.174	1.353
Full Coverage Previous Quarter	27.715	0.197	0.000	12.826	28.038
Child Age	−0.038	0.026	0.143	−0.080	0.005
Sibling Count	−1.343	0.074	0.000	13.826	−1.221
Social Vulnerability Index	−4.367	0.308	0.000	−4.872	−3.861
Zip Code is Urban	−2.478	0.205	0.000	−2.814	−2.142
*Panel D: New Congestive Heart Failure Diagnosis (n = 266)*					
New CHF DX	0.834	3.900	0.831	−5.561	7.230
Full Coverage Previous Quarter	27.722	0.197	0.000	27.398	28.045
Child Age	−0.042	0.026	0.105	−0.084	0.001
Sibling Count	−1.345	0.074	0.000	−1.467	−1.223
Social Vulnerability Index	−4.343	0.308	0.000	−4.848	−3.837
Zip Code is Urban	−2.469	0.205	0.000	−2.806	−2.133

Notes: There are 1,101,657 quarters of child data, or approximately 7.05 quarters per child. Coefficient estimates are from a child fixed effects model that also included all of the variables shown in the table. Confidence limits (CL) are for a 95% confidence interval. Each panel shows results from a separate model.

## Data Availability

The data used in this study is proprietary private health insurance claims data which is not publicly available. The authors can however, make their programs available so that the study can be replicated in other similar data.

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
