# Peer review of "New Caregiver Diagnoses of Severe Depression and Child Asthma Controller Medication Adherence"

_ijerph, 2023, doi:10.3390/ijerph20115986_

Round 1
Reviewer 1 Report
The paper addresses a critically important and understudied area in pediatrics: the effects of serious adult (parent/caregiver) health conditions on child health conditions. The particular focus on adult depression and child asthma medication refills is apt given each is highly prevalent and functionally impactful. The large sample size and fixed effects model used to study the question are major advances over the prior literature. There are minor considerations offered to increase the clarity of the methods and to illuminate the potential mechanisms at play.
Title: Appropriate (see comment below re: use of the modifier “severe”)
Abstract:
· Clear and appropriate
· For clarity consider changing second sentence to “But it is less clear how adherence responds to a caregiver’s new diagnosis of severe depression…” to make clear the change is the new diagnosis not a new caregiver.
Background
- Provides appropriate scientific context for the research question and is clearly written
- I would recommend adding a sentence to sharpen the critique of prior observational studies to highlight the value of this study’s fixed effects modeling.
Methods
- Overall study design and methods are excellent and laid out clearly.
- Some additional comments/considerations:
o Spell out BHI in its first use
o To better understand generalizability (while respecting business confidentiality), can anything further be said about what commercial plans are included in BHI’s database?
o The actual asthma diagnosis codes could be clearer as to whether wheeze is included or not, especially since asthma isn’t a definitive diagnosis at their lower age limit of 2 years.
o It would help to spell out the outcome more clearly – when are the actual 90 days child medication windows in relation to the incident/new claim for CG depression? Is the outcome the change in # of days covered in the 1st quarter after depression claim compared to the quarter just before the 1st claim? Or averaged across all quarters after vs. before?
o It would also help to be clearer on the depression diagnosis – is there a modifier for ‘Severe’ or are these CG’s Major Depressive Disorder w/wo psychotic disorder – in which case I’m not sure I would use the word ‘severe’ throughout the paper. (Perhaps “serious” as an alternative?) An adult mental health services researcher would be best positioned to answer that
o Given the proposed mechanism of action between CG depression and medication adherence, one might also look for a potential interaction between child age and CG depression – ie one could hypothesize a greater effect of CG depression on younger children (eg 2-7 vs. 7-11 years)
o It seems like parental age might be a relevant variable of interest
o There were an average of 7.05 quarters of observations. How many quarters of observation before and after new caregiver dx?
Results
- The study results are clearly written and relevant findings highlighted.
- Some additional comments/considerations:
o I think the biggest surprise for me is the low frequency of incident caregiver diagnoses for Depression and 10 fold higher number of diagnoses for COPD (619 vs. 6958 if I’m reading the tables correctly). That seems very strange for the age cohort of parents of children 2-11 years old (especially as noted in the Background that 18% of adults have depressive symptoms). It is less surprising if there is a specific claims modifier to only include SEVERE depression.
Discussion
- The Discussion and Conclusion are appropriate to the research question, findings and broader context. The advantages of this study design are important to note and are clearly laid out.
- Additional comments/considerations
o I recommend 1-2 sentences at the beginning of the Discussion summarizing the results
o While 2 potential confounding explanations are ruled out in the 1st paragraph, I would suggest a sentence or two on what might be the mechanisms that explains the findings. How might an incident serious condition in the Caregiver affect prescription refills for the child?
o Potential explanations for why there are significant results for Diabetes and Depression but not Cancer or others would also help the reader (eg is there greater mismeasurement? The time lags for parenting impact are different? Different early burdens on parental time and finances?)
Limitations
- I would add that this data source is only for children with private insurance and the generalizability to those with public insurance needs further examination
Conclusion
Either for the Discussion or Conclusion – Support for the Caregiver and follow up are important considerations. I might add an additional consideration is to ensure child medication regimens are simplified (minimize the number of medications and frequency of) and easily accessible (e.g., home delivery or school nurse delivered when possible).
Table 1
- For child gender and urban zip code, it seems it would be better to report the % of a given category in columns 1 & 2 instead of the N
References
- Appropriate
Author Response
Response to Reviewer #1
Thank you for your thoughtful comments which have helped us to improve the paper. We have included the original comment in italics and our response in plain text.
The paper addresses a critically important and understudied area in pediatrics: the effects of serious adult (parent/caregiver) health conditions on child health conditions. The particular focus on adult depression and child asthma medication refills is apt given each is highly prevalent and functionally impactful. The large sample size and fixed effects model used to study the question are major advances over the prior literature. There are minor considerations offered to increase the clarity of the methods and to illuminate the potential mechanisms at play.
Title: Appropriate (see comment below re: use of the modifier “severe”)
Abstract:
- Clear and appropriate
- For clarity consider changing second sentence to “But it is less clear how adherence responds to a caregiver’s new diagnosis of severe depression…” to make clear the change is the new diagnosis not a new caregiver.
We have made this change.
Background
- Provides appropriate scientific context for the research question and is clearly written
- I would recommend adding a sentence to sharpen the critique of prior observational studies to highlight the value of this study’s fixed effects modeling.
We have added the following sentence at the end of the Introduction, just before the Methods section:
“This study differs from previous work because the data follows the same child before and after the parent’s diagnosis in order to observe the changes rather than only observing a correlation between medication adherence and the parent’s mental health condition.”
Methods
- Overall study design and methods are excellent and laid out clearly.
- Some additional comments/considerations:
o Spell out BHI in its first use
Done
To better understand generalizability (while respecting business confidentiality), can anything further be said about what commercial plans are included in BHI’s database?
We have added the following explanation: “This study used administrative claims from Blue Health Intelligence (BHI), a licensee of the Blue Cross Blue Shield Association. The dataset includes enrollment information, medical claims, and pharmacy claims for Blue Cross and Blue Shield plan beneficiaries across the country. It does not include Medicaid nor Medicare Advantage.”
The actual asthma diagnosis codes could be clearer as to whether wheeze is included or not, especially since asthma isn’t a definitive diagnosis at their lower age limit of 2 years.
We have added a list of diagnosis codes to the supplemental materials as Table S2. Aside from an asthma ICD10 code we also required subsequent treatment with one of the medications listed, a restriction which is now clarified in the text.
It would help to spell out the outcome more clearly – when are the actual 90 days child medication windows in relation to the incident/new claim for CG depression? Is the outcome the change in # of days covered in the 1st quarter after a depression claim compared to the quarter just before the 1st claim? Or averaged across all quarters after vs. before?
We are defining all of the variables with respect to calendar year quarters. For each quarter, we define the medication possession ratio (our measure of adherence) as the number of days supply divided by the number of days in the period. Because we are estimating child fixed effects models, we are essentially looking at the change in this ratio for all quarters before the new diagnosis compared to the quarter in which the caregiver receives the new diagnosis and the following quarters. We have tried to clarify this in the text.
It would also help to be clearer on the depression diagnosis – is there a modifier for ‘Severe’ or are these CG’s Major Depressive Disorder w/wo psychotic disorder – in which case I’m not sure I would use the word ‘severe’ throughout the paper. (Perhaps “serious” as an alternative?) An adult mental health services researcher would be best positioned to answer that.
We have clarified that we only used ICD10 codes with a description of “severe” depression. We also list the codes which are F322, F323, F332 and F333, each of which includes the modifier “severe.”
Given the proposed mechanism of action between CG depression and medication adherence, one might also look for a potential interaction between child age and CG depression – ie one could hypothesize a greater effect of CG depression on younger children (eg 2-7 vs. 7-11 years).
We do include child age as a regressor in the model and it is significantly negative (i.e. older children are less likely to be adherent to their medication regime. However, splitting the sample by age results in two samples of about half the size and larger standard errors on the key variables of interest such that we cannot say with any confidence whether the effects of parental diagnoses are bigger or smaller for younger children.
It seems like parental age might be a relevant variable of interest
We now clarify that parental age at the start of the sample is one of the things that is captured by the child fixed effect, and that the change in child age is identical to the change in parent age over the time we observe the child.
There were an average of 7.05 quarters of observations. How many quarters of observation before and after new caregiver dx?
On average we see 2.2 quarters post diagnosis and 8.2 quarters pre-diagnosis. We have added this information to the text.
Results
- The study results are clearly written and relevant findings highlighted.
- Some additional comments/considerations:
I think the biggest surprise for me is the low frequency of incident caregiver diagnoses for
Depression and 10 fold higher number of diagnoses for COPD (619 vs. 6958 if I’m reading the
tables correctly). That seems very strange for the age cohort of parents of children 2-11 years old
(especially as noted in the Background that 18% of adults have depressive symptoms). It is less
surprising if there is a specific claims modifier to only include SEVERE depression.
Yes, the included list of ICD codes only include ‘severe’ depression as we have tried to clarify in the text.
Discussion
- The Discussion and Conclusion are appropriate to the research question, findings and broader context. The advantages of this study design are important to note and are clearly laid out.
- Additional comments/considerations
I recommend 1-2 sentences at the beginning of the Discussion summarizing the results
Thank you for suggesting this. We have added a summary sentence.
While 2 potential confounding explanations are ruled out in the 1st paragraph, I would suggest a
sentence or two on what might be the mechanisms that explains the findings. How might an incident
serious condition in the Caregiver affect prescription refills for the child?
Thank you for this suggestion. We added this explanation at the start of this section:
“There are many reasons that a new diagnosis of severe depression in a caregiver might be associated with reduced medication adherence among children. The caregiver might find that the new demands associated with their own treatment (such as additional therapy) crowd out the time available for dealing with their child’s medical problems. Or they might find the diagnosis itself overwhelming so that they have more difficulty dealing with other demands on their attention. The depression itself might also make it more difficult for the caregiver to attend to the needs of the child.”
Potential explanations for why there are significant results for Diabetes and Depression but not Cancer or others would also help the reader (eg is there greater mismeasurement? The time lags for parenting impact are different? Different early burdens on parental time and finances?)
We have noted this question but did not want to speculate too much about the possible causes. One possibility is that people with new cancer, COPD, and CHF diagnoses initially receive more sympathy and support than those with severe depression and diabetes, which might help them to continue to look after others in their families. We have added references suggesting that there may be special stigma associated with depression and diabetes.
Limitations
- I would add that this data source is only for children with private insurance and the generalizability to those with public insurance needs further examination
We have added this limitation.
Conclusion
Either for the Discussion or Conclusion – Support for the Caregiver and follow up are important considerations. I might add an additional consideration is to ensure child medication regimens are simplified (minimize the number of medications and frequency of) and easily accessible (e.g., home delivery or school nurse delivered when possible).
We have added these points to the conclusion, thank you.
Table 1
- For child gender and urban zip code, it seems it would be better to report the % of a given category in columns 1 & 2 instead of the N
We have added them.
References
- Appropriate

Reviewer 2 Report
The topic of the manuscript is very interesting and it is needed in the literature. I recommend publish this manuscript taking into consideration the following changes:
Introduction: I suggest expand the introduction and focus on others studies. The authors could summaries their design, sample size, and the outcome in table.
Methods: Could the authors add clear scientific justification regarding not getting the approval by Institutional Review Board.
I suggest use “the authors” or “the study” instead of “we” in the manuscript.
Please add more information about the type of statistical analyses (t-test, ANOVA, etc.)
The abbreviations “P” should be italic throughout the manuscript because it represents a number.
Language and punctuation: delete the space line 39, 61, 65, 86, 115, 139, 151, 156, 164, 172, 175
In table 1 clarify what do you mean by difference 2-1 (column 3)
I suggest rewrite mean ± standard deviation for the values e.g. as 4.84 ± 2.39 instead of 4.84 (2.39)
11 references is not accepted for such study.
Author Response
Response to Reviewer 2
Thank you for your thoughtful comments which have helped us to improve the paper. We have included the original comment in italics and our response in plain text.
The topic of the manuscript is very interesting and it is needed in the literature. I recommend publish this manuscript taking into consideration the following changes:
Introduction: I suggest expand the introduction and focus on others studies. The authors could summaries their design, sample size, and the outcome in table.
We have added more discussion and a summary table as Table S4.
Methods: Could the authors add clear scientific justification regarding not getting the approval by Institutional Review Board.
We apologize for not being clearer that the data set we use is completely de-identified so that what we are doing is not human subjects research and does not require IRB approval. We have clarified this point in the manuscript.
I suggest use “the authors” or “the study” instead of “we” in the manuscript.
The study has been edited to eliminate “we.”
Please add more information about the type of statistical analyses (t-test, ANOVA, etc.)
We have added this information.
The abbreviations “P” should be italic throughout the manuscript because it represents a number.
Done.
Language and punctuation: delete the space line 39, 61, 65, 86, 115, 139, 151, 156, 164, 172, 175
Apologies, but we could not find the spaces that you are referencing. We hope the copy editor will adjust the text if there is a problem with spacing.
In table 1 clarify what do you mean by difference 2-1 (column 3)
We have clarified that it is the difference in the means between the two groups of children.
I suggest rewrite mean ± standard deviation for the values e.g. as 4.84 ± 2.39 instead of 4.84 (2.39)
Done.
11 references is not accepted for such study.
We have added additional references and now have 26.
Reviewer 3 Report
Dear authors,
Brief summary
This manuscript was designed with the aim to assess the adherence to the prescription refill and the effect of the severe depression of the caregivers towards the refill time.
General comment
Introduction
· It would be good if the author described more on the uniqueness of this study. What are the research gaps from the previous studies and how the current work in this manuscript has addressed the gaps? This explanation will increase the weightage of this manuscript to be considered for publication.
· It would be good if the author described on how to determine the medication adherence, as there are many methods to measure medication adherence. The authors may also add the description on the use of prescription filling to measure the medication adherence.
Method
· Ethical approval: Usually, all study involving human data need an ethical approval from the relevant bodies, however, this may vary from country to country. It would be good if the author could add the references (eq. from the local research ethics guideline) stating that no ethical approval needed when retrospective human data were used in a study.
· Sampling method is needed. How were the subjects chosen? Purposive sampling? Convenience sampling?
· It would be good if the author included the references on the definition of medication adherence in this manuscript. (Medication adherence was measured as the number of days in the previous days that a child was covered by at least one asthma controller medication.)
· How was the level of adherence determined? What was the cut off points for the number of days the patients took the medication?
· It would be more informative if the author described or summarized, with references, the STROBE guideline
Results
· Table 1: It would be good if the author described on to what extent of every significant factors contributed to the development of depression.
· Figure 1 containing abbreviation that was not defines as footnote. It would be good if the authors described more on the effects of depression and other newly diagnosed disease on the medication adherence, ie to what extent are the effects?
· There are no clear results on the level of medication adherence in the study, as implied in the title. It would be good if the authors add some results on the level of medication compliance of these cohort of patients.
Discussion
· The discussion should be more thorough. Every effect and factor (especially the significant ones) should be explained and supported by the references from the literature. Critical evaluation should also be included in the discussion as well.
Author Response
Response to Reviewer 3
Thank you for your thoughtful comments which have helped us to improve the paper. We have included the original comment in italics and our response in plain text.
Introduction
It would be good if the author described more on the uniqueness of this study. What are the research gaps from the previous studies and how the current work in this manuscript has addressed the gaps? This explanation will increase the weightage of this manuscript to be considered for publication.
Thank you for this suggestion. We have added some additional explanation especially focusing on the fact that child illness could cause parental depression rather than vice versa. We now explain that this is why it is important to be able to follow children over time, before and after their parent’s diagnosis.
It would be good if the author described on how to determine the medication adherence, as there are many methods to measure medication adherence. The authors may also add the description on the use of prescription filling to measure the medication adherence.
We have clarified the definition of medication adherence that we are using as follows: “The measure of medication adherence is the medication possession ratio, which is equal to the number of days supply (of one of the listed medications) divided by the number of days in the period. This variable is defined with reference to calendar quarters. This measure is commonly employed by researchers using administrative pharmacy claims databases.18 Using pharmacy claims the best one can say is that the subscription was filled and that the patient had enough supply to cover the days in the quarter. This is a minimum criterion for medication compliance, but whether the child actually took the medication as prescribed is not observed in these data.”
We have also added a reference supporting the use of this measure.
Method
Ethical approval: Usually, all study involving human data need an ethical approval from the relevant bodies, however, this may vary from country to country. It would be good if the author could add the references (eq. from the local research ethics guideline) stating that no ethical approval needed when retrospective human data were used in a study.
We are sorry to have caused confusion about this important issue. We now clarify that the data in the BHI warehouse are completely de-identified, so that they are not human subjects data.
Sampling method is needed. How were the subjects chosen? Purposive sampling? Convenience sampling?
We are not sampling. We are using all of the children in the BHI data warehouse who meet the study criterion. We have tried to clarify this important issue in the manuscript. We have edited to get rid of the word “sample” in order to avoid confusion that we may have inadvertently created.
It would be good if the author included the references on the definition of medication adherence in this manuscript. (Medication adherence was measured as the number of days in the previous days that a child was covered by at least one asthma controller medication.)
How was the level of adherence determined? What was the cut off points for the number of days the patients took the medication?
Please see our response above about the definition of medication adherence which we have now included in the paper.
We also state that a limitation of the study is that we cannot observe whether a child actually took the medication.
And we have added a reference to a study supporting the use of this definition of medication adherence.
It would be more informative if the author described or summarized, with references, the STROBE guideline.
We have added a reference to the STROBE guideline and clarified that it is a checklist that some medical journals use to make sure an author has covered the main points.
Results
Table 1: It would be good if the author described on to what extent of every significant factors contributed to the development of depression.
We apologize if we conveyed the impression that Table 1 was supposed to explain the development of depression. The purpose of Table 1 is to show that families whose caregivers have new diagnoses of depression are different from other families in ways that might also affect their children’s medication compliance. This is why it is important to follow the same child over time. We have tried to clarify this important point.
Figure 1 containing abbreviation that was not defines as footnote. It would be good if the authors described more on the effects of depression and other newly diagnosed disease on the medication adherence, i.e. to what extent are the effects?
We have written out the abbreviations.
There are no clear results on the level of medication adherence in the study, as implied in the title. It would be good if the authors add some results on the level of medication compliance of these cohort of patients.
Tables 2 and 3 present the effects of a new caregiver diagnosis of depression on the indicator of medication compliance that we are using. We apologize that this was not clear and have added the following explanation:
“Table 2 shows the estimated impact of a new caregiver diagnosis of severe depression on medication adherence. That is, the table shows the results of estimating regression models in which medication adherence is the dependent variable and the independent variables include child age, an indicator equal to one if the child had full medication coverage in the previous period and zero otherwise, the number of siblings, an indicator equal to one if the postal code is urban and zero otherwise, the SVI, which is standardized to have a mean of zero and a standard deviation of one, and a zero-one indicator for each child (i.e. the child fixed effect).”
Discussion
The discussion should be more thorough. Every effect and factor (especially the significant ones) should be explained and supported by the references from the literature. Critical evaluation should also be included in the discussion as well.
We have added more discussion and further references.
Round 2
Reviewer 2 Report
All comments were done.
Reviewer 3 Report
Dear authors,
My previous comments have been adequately addressed.